# Site-specific anisotropic assembly of amorphous mesoporous subunits on crystalline metal–organic framework

Minchao Liu[1], Cheng Shang [1], Tiancong Zhao [1], Hongyue Yu[1], Yufang Kou[1], Zirui Lv[1], Mengmeng Hou[1], Fan Zhang [1], Qiaowei Li [1], Dongyuan Zhao [1] & Xiaomin Li [1] ✉

As an important branch of anisotropic nanohybrids (ANHs) with multiple surfaces and functions, the porous ANHs (p-ANHs) have attracted extensive attentions because of the unique characteristics of high surface area, tunable pore structures and controllable framework compositions, etc. However, due to the large surface-chemistry and lattice mismatches between the crystalline and amorphous porous nanomaterials, the site-specific anisotropic assembly of amorphous subunits on crystalline host is challenging. Here, we report a selective occupation strategy to achieve site-specific anisotropic growth of amorphous mesoporous subunits on crystalline metal–organic framework (MOF). The amorphous polydopamine (mPDA) building blocks can be controllably grown on the {100} (type 1) or {110} (type 2) facets of crystalline ZIF-8 to form the binary super-structured p-ANHs. Based on the secondary epitaxial growth of tertiary MOF building blocks on type 1 and 2 nanostructures, the ternary p-ANHs with controllable compositions and architectures are also rationally synthesized (type 3 and 4). These intricate and unprecedented superstructures provide a good platform for the construction of nanocomposites with multiple functionalities and understanding of the structure-property-function relationships.

Anisotropic nanohybrids (ANHs) are a class of emerging materials with asymmetric structure and multiple surface chemistries, in which two or more distinctive subunits are anisotropically fused together[1–5]. They can offer unlimited possibilities in either promoting properties of their individual subunits or integrating various functional components within one single structure, even generating desirable properties and functions[6]. The fascinating properties and functions of the ANHs depend on not only its composition[7–9], but also its structure[10,11], exposed facet[12,13], spatial distributions of each subunit[14] and the interface between the subunits[15]. Therefore, the rational design and controlled synthesis of ANHs is of importance not only for the

realization of improved application performance, but also for enhancing our understanding of the fundamental structure-property-function relationships.

Among the various building blocks of ANHs, the porous nanoparticles with high surface area, tunable pore sizes and structures, controllable framework compositions[16–19], etc. are becoming an exciting candidate for the construction of ANHs. Although it is still in its infancy, the porous ANHs (p-ANHs) with multiple porous building blocks have been explored after a few years of rapid development, including crystalline p-ANHs based on crystalline metal–organic frameworks (MOFs)[20–24] and amorphous p-ANHs based on the amorphous

[1]Department of Chemistry, Shanghai Stomatological Hospital & School of Stomatology, State Key Laboratory of Molecular Engineering of Polymers, iChem (Collaborative Innovation Center of Chemistry for Energy Materials), Shanghai Key Laboratory of Molecular Catalysis and Innovative Materials, Fudan University, 200433 Shanghai, China. ✉e-mail: lixm@fudan.edu.cn

mesoporous silica, polymer and carbon[25–29]. The p-ANHs can not only provide multiple independent surfaces and unique heterojunctions for the enhanced matter/energy exchange efficiency with external environments, but also possess anisotropic storge spaces for site-specific loading of functional guests[30–32]. These features make them have great application prospects in multi-guests co-delivery[29], nanomedicine[30], biphasic cascade catalysis[31,32], photocatalysis[20], and so on.

The facet-specific epitaxial growth of crystalline microporous building blocks can be easily realized based on the lattice matching configurations (e.g., MOF on MOF p-ANHs)[20–24]. However, due to the amorphous feature of most mesoporous nanomaterials, the surface chemistry and lattice mismatching at the interface between the crystalline and amorphous subunits are very large, and the anisotropic growth sites of amorphous mesoporous subunit on the host are usually non-specific[25–29]. So, in a long time, the framework of the obtained p-ANHs is limited to the amorphous-on-amorphous or crystalline-on-crystalline combination. There is an obvious research "boundary" between the two types of p-ANHs. The site-specific anisotropic growth of amorphous porous subunits on crystalline porous building blocks is challenging, but urgently desired for the extension of p-ANHs with controllable architectural, compositional and functional complexity.

In this paper, we develop the selective occupation strategy to achieve the controllable site-specific anisotropic growth of amorphous mesoporous polydopamine (mPDA) on crystalline ZIF-8 nanoparticles. By regulating the interface between the crystalline and amorphous subunits, four types of p-ANHs with intricate and unprecedented architectures is reported, and the research "boundary" between the crystalline p-ANHs and amorphous p-ANHs is crossed. Owing to the different coordination environment of zinc ions on the {100} and {110} facets of truncated rhombic dodecahedral (TRD) ZIF-8 nanoparticle, the mPDA nanoplates can be selectively assembled on the six {100} facets of crystalline ZIF-8 (type 1). In addition, through the selective occultation strategy of small molecules, the coordination sites on {100} planes are occupied, leading to the anisotropic growth of mPDA nanoplates on the {110} planes of ZIF-8 host (type 2). Based on the secondary epitaxial growth of crystalline MOF, the exposed facets on the obtained type 1 and 2 binary p-ANHs can be further rationally arranged the growth of tertiary MOF building blocks with varied components to form intricate and unprecedented ternary anisotropic hybrid superstructures (type 3 and 4).

## Results and discussion

As one of the most typical representatives of crystalline MOFs, the monodispersed ZIF-8 nanoparticles were synthesized[33] and used as crystalline host for the following anisotropic assembly of amorphous mesoporous subunit. As show in Supplementary Fig. 1, the diameter of the ZIF-8 nanoparticles (the edge length of equivalent cube) can be well tuned from 200 to 300 nm. The regular TRD morphology with 12 {110} and 6 {100} facets of the nanoparticles can be clearly observed. The X-ray diffraction (XRD) patterns further confirms the cubic crystalline nature ($I\bar{4}3m$) of the ZIF-8 nanoparticles (Supplementary Fig. 2). The TRD ZIF-8 with the diameter of ~260 nm was used as a typical pristine host for the following site-specific assembly of amorphous mesoporous subunits.

### Type 1 binary-superstructure

As shown in Fig. 1A, the type 1 p-ANHs are synthesized in an emulsion system formed by using H₂O/ethanol (6:4) solution as aqueous phase, 2,3,6-trimethylphenol (TMP)/1,3,5-trimethylbenzene (TMB) as oil phase, Pluronic® F-127 as both surfactants for the emulsion stabilization and mesostructure-directing agent for the formation of mesopores in the mPDA. The TRD ZIF-8 nanoparticles are used as host for the deposition of dopamine precursors under alkaline conditions (pH = ~10). In this wet chemistry synthesis, because of the different

coordination environment of $Zn^{2+}$ on {100} and {110} facets of ZIF-8 nanoparticle, six amorphous plate-shaped mPDA patches are selectively grown on {100} facets of the crystalline TRD ZIF-8 host to form the type 1 p-ANHs.

The scanning electron microscope (SEM) and transmission electron microscope (TEM) images clearly show the anisotropic and hybrid structure of type 1 p-ANHs (Fig. 1B, C and Supplementary Fig. 3), which are composed of six mPDA nanoplates with vertical mesopore channels and pristine TRD ZIF-8 host. The high-angle annular dark-field scanning TEM (HAADF-STEM) image of type 1 p-ANHs clearly shows that the brightness of central square area is higher (marked with red line), and the surrounding patches are lower (marked with yellow line), corresponding to the metal-enriched ZIF-8 and mPDA subunits, respectively (Fig. 1D). The diameter of ZIF-8 building block remained almost constant at about 260 nm after the selective growth of mPDA subunits on its {100} facets. The average thickness of mPDA nanoplates is about 40–50 nm, and the pore size of the vertical mesopore channels is about 20 nm. The asymmetric geometry and element distributions of the obtained type 1 p-ANHs can be further confirmed by the energy-dispersive X-ray spectroscopy (EDX) elemental mapping (Fig. 1E). All the expected elements, including Zn (from the ZIF-8), C and N (from both the linker of ZIF-8 and mPDA) can be detected and match well with the model of type 1 architecture. Zn element can also be clearly observed in the mPDA subunits, indicating that the $Zn^{2+}$ in the pristine ZIF-8 migrated from the MOF framework to the mPDA subunits. We speculate that this migration is induced by the strong coordination effect of dopamine with $Zn^{2+}$, and this migration may cause some defects at the interface of mPDA and ZIF-8. After etching the MOF building blocks with acid, uniform square slices of mPDA are obtained (Supplementary Fig. 4), indicating that the mPDA subunits on the MOF host are isolated from each other without cross-linking, and further proving the exposure of {110} facets in the type 1 architecture.

The crystalline structure of type 1 p-ANHs can be determined by TEM images and the corresponding selected-area electron diffraction (SAED) patterns of a single-particle from different perspectives (Fig. 1F). The projected contours of the nanoparticles fit well with the corresponding model of type 1 architecture. Along the zone axis of [100], [110], and [111], the SAED patterns are consistent with single-crystal structure and orientations of TRD ZIF-8 host. These results strongly suggest the selective growth of amorphous mPDA nanoplates on {100} facets of crystalline TRD ZIF-8 host. The XRD patterns of the obtained type 1 p-ANHs show identical diffraction peaks as pristine ZIF-8 nanoparticles, conforming the good crystallinity of ZIF-8 building blocks in the nanohybrids (Supplementary Fig. 2). In addition, compared to the pristine TRD ZIF-8, the diffraction peak intensity ratio between {002} and {022} facets remained unchanged after selective growth of mPDA subunits, indicating that the selective growth of amorphous mPDA subunits neither affects the macroscopic morphology of pristine ZIF-8 host, nor affects its microscopic crystal structure.

Nitrogen sorption isotherm of type 1 p-ANHs shows the combined type I and type IV isotherms with obvious nitrogen adsorption capacity at relative pressures of 0–0.05 and 0.70–0.95 (Fig. 1G), indicating the presence of micropores and mesopores in the nanohybrids. The pore size distribution curve shows a narrow peak at about 0.8 nm and a broad peak at 15–60 nm, which can be attributed to the micropores in the ZIF-8 building blocks and the mesopores in the mPDA nanoplates, respectively. By removing the ZIF-8 building blocks with acid, we can calculate the mass proportion of mPDA building blocks in type 1 p-ANHs as 47.6 %. Based on this, the calculated net Brunauer-Emmett-Teller (BET) surface area of ZIF-8 building blocks in type 1 p-ANHs is 1086 m²/g, which is smaller than the net surface area of the pristine ZIF-8 nanoparticles (1457 m²/g) (Supplementary Table 1). We speculate that the growth of mPDA on the {100} facets of pristine ZIF-8 may cause the blockage of micropores on the {100} facets of the pristine ZIF-8, which further result in the decrease of the specific surface of the

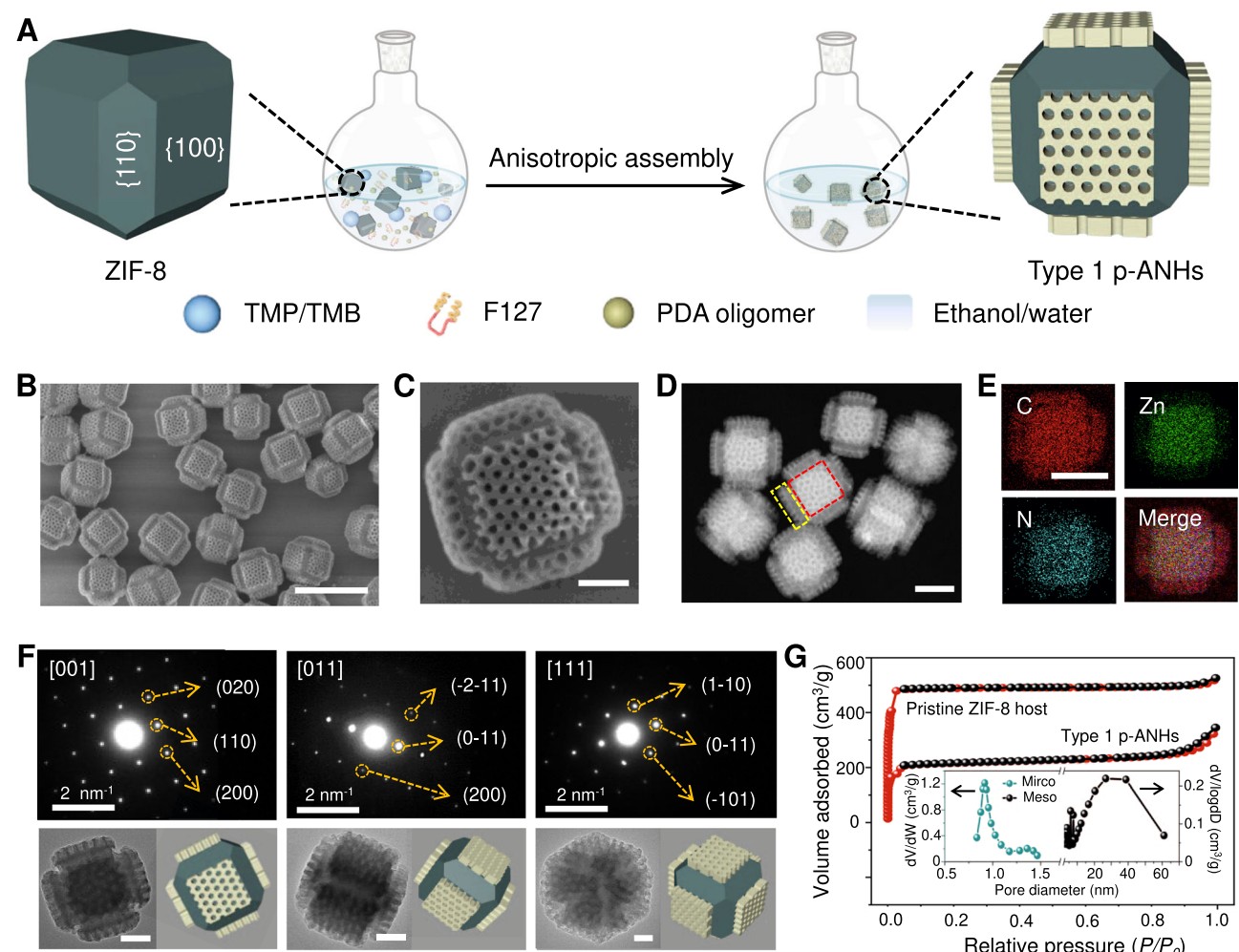

**Fig. 1 | Synthesis and characterizations of type 1 p-ANHs. A** Schematic illustration of the synthesis process. **B**, **C** SEM images with different magnifications, **D** HAADF-STEM image and **E** EDX elemental mapping of type 1 p-ANHs. **F** TEM images, corresponding models and SAED patterns taken from a single nanoparticle along [100], [110], and [111] zone axis. **G** Nitrogen sorption isotherms of the pristine ZIF-8 host and type 1 p-ANHs (inset: the pore size distribution of type 1 p-ANHs). Scale bars: 500 nm for **B**, 100 nm for **C**, 200 nm for **D**, 200 nm for **E**, 100 nm for **F**. Source data are provided as a Source Data file.

micropores. In addition, the defects at the interface of mPDA and ZIF-8 may also induce the decrease of surface area of pristine ZIF-8.

The diameter of type 1 p-ANHs can be well tuned by controlling the size of the pristine ZIF-8 host (Supplementary Fig. 5). The precise manipulation of the mesopore size of mPDA subunits can be achieved by varying the ratio of TMP/TMB in the oil phase (Supplementary Fig. 6). For example, without the addition of TMP, the pore size is about 5 nm, which can be further increased to 10 and 20 nm as the increase of TMP in the oil phase to 150 and 200 mg, respectively. By tuning the concentration of the dopamine precursor, the thickness of the mPDA nanoplates can be adjusted in the range from 15 to 70 nm (Supplementary Fig. 7). In addition, by using the TRD ZIF-8 with different {110}/{100} exposure ratio as pristine host, the selective growth of mPDA subunits can still be carried out (Supplementary Figs. 8 and 9), which demonstrates the university of the synthesis method and provides convenience for us to regulate the proportion of mesopores and micropores in the type 1 p-ANHs. Moreover, the anisotropic growth of mPDA can also be applied to titanium MOF of MIL-125, and amorphous $mSiO_2$ nanoplates can also be anisotropiccally grown on the {100} facets of TRD ZIF-8 (Supplementary Figs. 10 and 11).

### Type 2 binary-superstructure
In a similar emulsion system, another distinct p-ANHs (type 2) can be synthesized by increasing $H_2O$/ethanol ratio and TMP amount in

the oil phase to 8:2 and >200 mg, respectively (Fig. 2A). In this synthesis conditions, the coordination sites on {100} planes are occupied by TMP small molecules. So, instead of selective growth on {100} facets, the mPDA subunits with vertical mesopores are selectively grown on the {110} facets of the TRD ZIF-8 host. As shown in the large-area SEM and TEM images (Fig. 2B and Supplementary Fig. 12), the type 2 p-ANHs with diameter of ~300 nm inherit the uniform morphology of the pristine ZIF-8 host. Six cylindrical canyons can be clearly observed on the {100} facets of each nanoparticle, indicating that there is no mPDA on these facets, which can also be confirmed by the HAADF-STEM image (Fig. 2C) and single-particle TEM images taken from different perspectives (Fig. 2D). The length, width and thickness of each mPDA nanoplate on the {110} facets are about 300, 100, and 30 nm, respectively. The mesopore size on the nanoplates is 15–40 nm, which is not as uniform as that of in the type 1 nanostructure. Moreover, the pore size of mPDA could be increased from 20 to 80 nm when the TMP in oil phase was increased from 200 to 300 mg (Supplementary Fig. 13). The width of the mPDA nanoplate can be varied by tuning the {110}/{100} exposure ratio on the pristine TRD ZIF-8 host (Supplementary Fig. 14).

The single-particle SAED (Fig. 2D) and XRD patterns (Supplementary Fig. 15) reveal the good crystallinity of type 2 p-ANHs and single-crystal nature of the ZIF-8 building block in each nanoparticle.

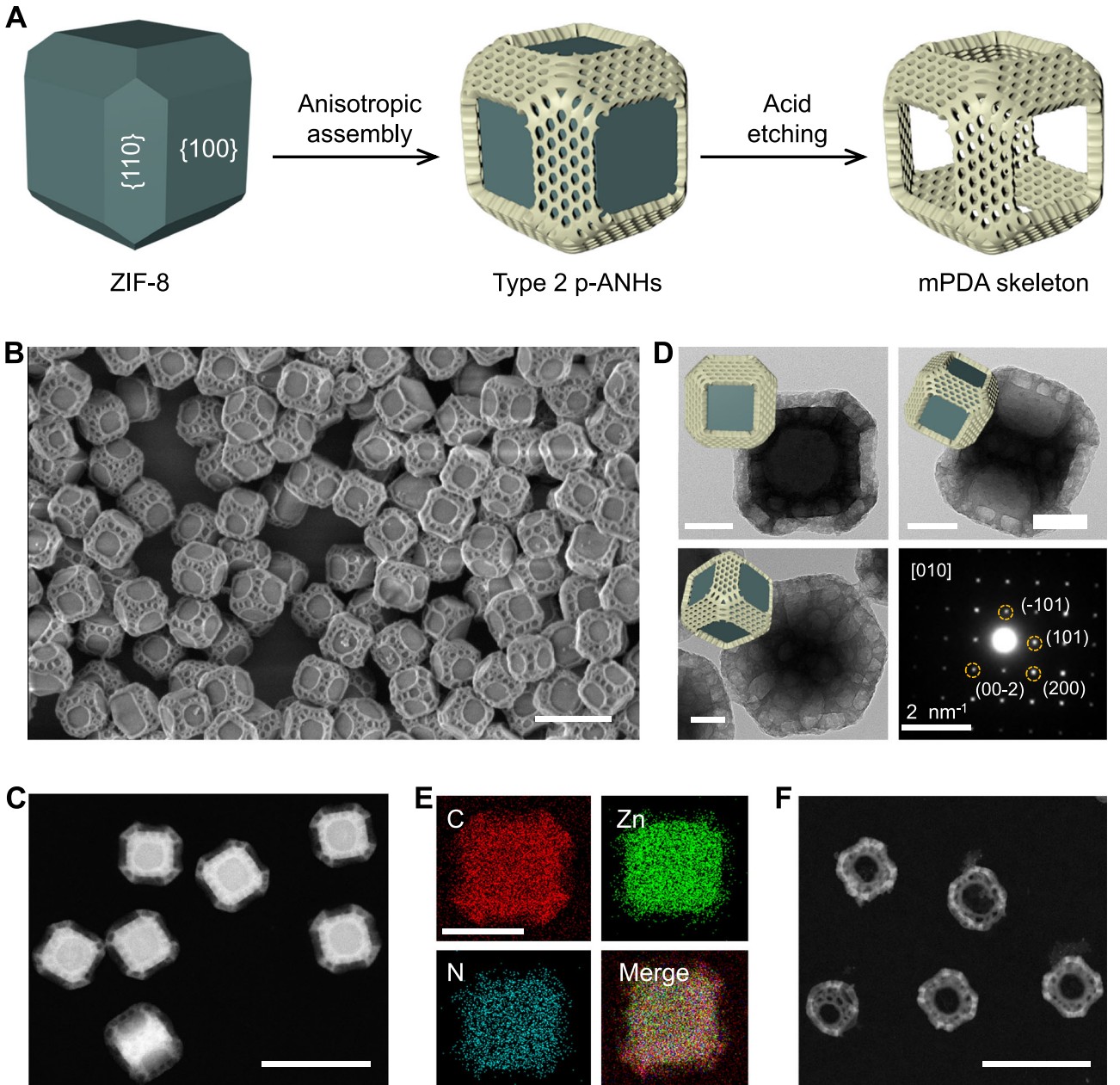

**Fig. 2 | Synthesis and characterizations of type 2 p-ANHs. A** Schematic illustration of the selective growth of mPDA nanoplates on {110} facets of TRD ZIF-8. **B** SEM and **C** HAADF-STEM images of type 2 p-ANHs. **D** TEM images of a single particle taken from different perspectives, and the corresponding SAED patterns taken from an entire nanoparticle along [010] zone axis. **E** EDX elemental mapping of type 2 p-ANHs. **F** HAADF-STEM images of the hollow cubical skeletons of mPDA. Scale bars: 500 nm for **B**, 500 nm for **C**, 100 nm for **D**, 200 nm for **E**, 200 nm for **F**.

The element distributions in the type 2 p-ANHs can be identified by EDX elemental mapping (Fig. 2E), further demonstrating the unique architecture of the nanohybrids. Since {110} facets on the TRD ZIF-8 are joined end to end, therefore, after etching the inner ZIF-8 building block from type 2 architecture, a complete hollow cubical skeleton can be obtained, which further demonstrates that the mPDA is selectively assembled on {110} facets (Fig. 2F and Supplementary Fig. 16). The mass proportion of mPDA subunits in type 2 p-ANHs is about 49.1 %. The calculate net BET surface area and pore volume of ZIF-8 building blocks and mPDA nanoplates are summarized in Supplementary Table 1 (Supplementary Fig. 17). Compared with the type 1 nanostructure, the micro- and meso-porosities of type 2 nanostructure do not change obviously, indicating that the selective growth of amorphous mPDA subunits on the {110} facets also do not influence the microporosity of pristine ZIF-8 host.

## Type 3 & 4 ternary-superstructures

Based on the aforementioned type 1 and 2 architectures, the intricate and unprecedented anisotropic hybrid ternary superstructures can be synthesized by the secondary epitaxial growth of hetero-MOF on the exposed facets.

Using type 1 p-ANHs as seeds, the tertiary building blocks of crystalline MOF can be epitaxially grown on the exposed {110} facets by directly immersing the seeds into the 2-methylimidazole and metal ions methanol solution (Fig. 3A). Here, we use ZIF-67 (Co²⁺) as the tertiary building blocks to fabricate the type 3 p-ANHs. Because of the low lattice mismatch, the tertiary building blocks of ZIF-67 can epitaxially grow on the exposed facets of pristine ZIF-8 host. SEM and TEM images of the obtained type 3 p-ANHs display a TRD morphology with six mPDA inlays (Fig. 3B and Supplementary Fig. 18). EDX elemental mapping and line scanning profile confirm the heterogeneous feature

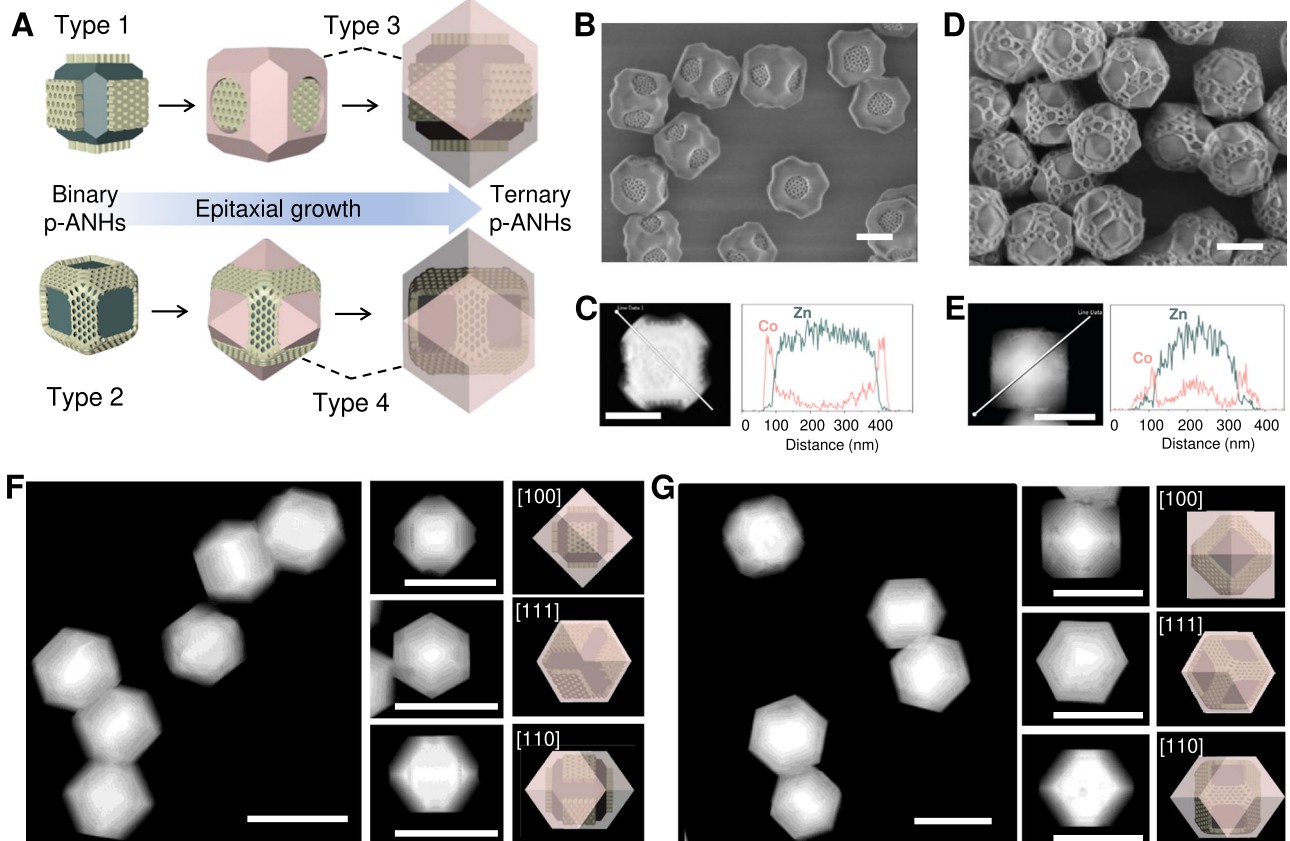

**Fig. 3 | Synthesis and characterizations of type 3 & 4 p-ANHs with ternary building blocks of ZIF-8, mPDA and ZIF-67. A** Schematic illustration of the formation of ternary p-ANHs of type 3 and 4 based on the epitaxial growth of crystalline ZIF-67. **B, C** SEM image and EDS line scanning profile of type 3 p-ANHs. **D, E** SEM image and EDS line scanning profile of 4 p-AHN. **F** HAADF-STEM images and the corresponding structure models of the rhombic dodecahedron type 3 p-ANHs with mPDA nanoplate inlays. **G** HAADF-STEM images and the corresponding structure models of the rhombic dodecahedron type 4 p-ANHs with mPDA cubical skeleton inlay. Scale bars: 200 nm for **B–E**, 500 nm for **F** and **G**. Source data are provided as a Source Data file.

of the type 3 p-ANHs, in which the spatial distribution of Co element is consistent with the structural model (Fig. 3C and Supplementary Fig. 18). The mPDA nanoplates inhibit the deposition of ZIF-67 precursors on {100} facets, leading to the orientation growth of tertiary ZIF-67 building blocks along [110] direction, which can be clearly evidenced by the gradually increased relative peak intensity of (011) in XRD patterns (compared with diffraction peak of (002)) (Supplementary Fig. 19). The epitaxial growth manner of the newly formed ZIF-67 building blocks on the {110} facets is also clearly evidenced by SAED patterns taken from an entire nanoparticle, which shows the typical diffraction spots of cubic single crystal.

Similarly, we can also realize the secondary epitaxial growth of ZIF-67 based on type 2 architecture, and the obtained ternary nanohybrids are labeled as type 4 (Fig. 3A). The SEM and TEM images, EDX mapping and line scanning profile show that six ZIF-67 pyramids are selectively grown on the exposed {100} facets of type 2 p-ANHs (Fig. 3D, E and Supplementary Fig. 20). Owing to the inhibition effect of mPDA on {110} facets, the morphology of the newly formed ZIF-67 building blocks on the exposed {100} facets gradually grow from quadrangular-frustum pyramid to rectangular pyramid (Supplementary Fig. 21). The SAED patterns taken from an entire nanoparticle reveal the single-crystalline nature of the obtained type 4 p-ANHs and the epitaxial growth manner of the tertiary ZIF-67 building blocks on the {100} facets (Supplementary Fig. 22).

With the continuous epitaxial growth of the exposed facets of crystalline ZIF-67, the inhibition effect of amorphous mPDA gradually weakened, resulting in the mPDA being gradually covered by tertiary ZIF-67 building blocks. Finally, the regular rhombic dodecahedron

nanoparticles are formed, which are composed by ternary building blocks of pristine ZIF-8, mPDA nanoplates or cubical skeleton inlays, outermost ZIF-67 (Fig. 3F, G). Beside the hetero-MOF of ZIF-67, the similar ternary superstructures can also be rationally synthesized by using the $Zn^{2+}$ based ZIF-8 as the tertiary building blocks (Supplementary Figs. 23–25). In addition, it worth noting that the growth of ZIF-67 in mesopore channels of mPDA subunits depends on the volume of added $Co^{2+}$ and 2-Methylimidazole (MeIM) solutions (Supplementary Figs. 26 and 27).

## The potential applications of the obtained type 1 p-ANHs
Unlike the core-shell structured nanohybrids obtained by isotropic growth/assembly strategy (which can only exhibit isotropic surface-chemistry of shell), one of the most important features of the obtained p-ANHs is that they can not only provide multiple independent and exposed surface chemistries for the enhanced matter/energy exchange efficiency with external environments, but also possess multiple surfaces/storage-spaces for site-specific loading/grafting of multiple functional entities. For example, the abundant micropores in the MOF subunits and mesopores in the mPDA subunits make the p-ANHs a good nanocarriers for multiple drugs co-delivery, and the excellent photothermal performance of mPDA subunits upon 808-nm near infrared (NIR) laser irradiation further endows p-ANHs with great potential as effective multifunctional therapeutic agent (Supplementary Figs. 29, 31, 32). As a proof of concept, the loading of cargoes, pH responsive drug release, photothermal performance and anti-tumor ability of the obtained type 1 p-ANHs were investigated. Type 1 p-ANHs were used as a model for the co-loading of disulfiram (DSF) in

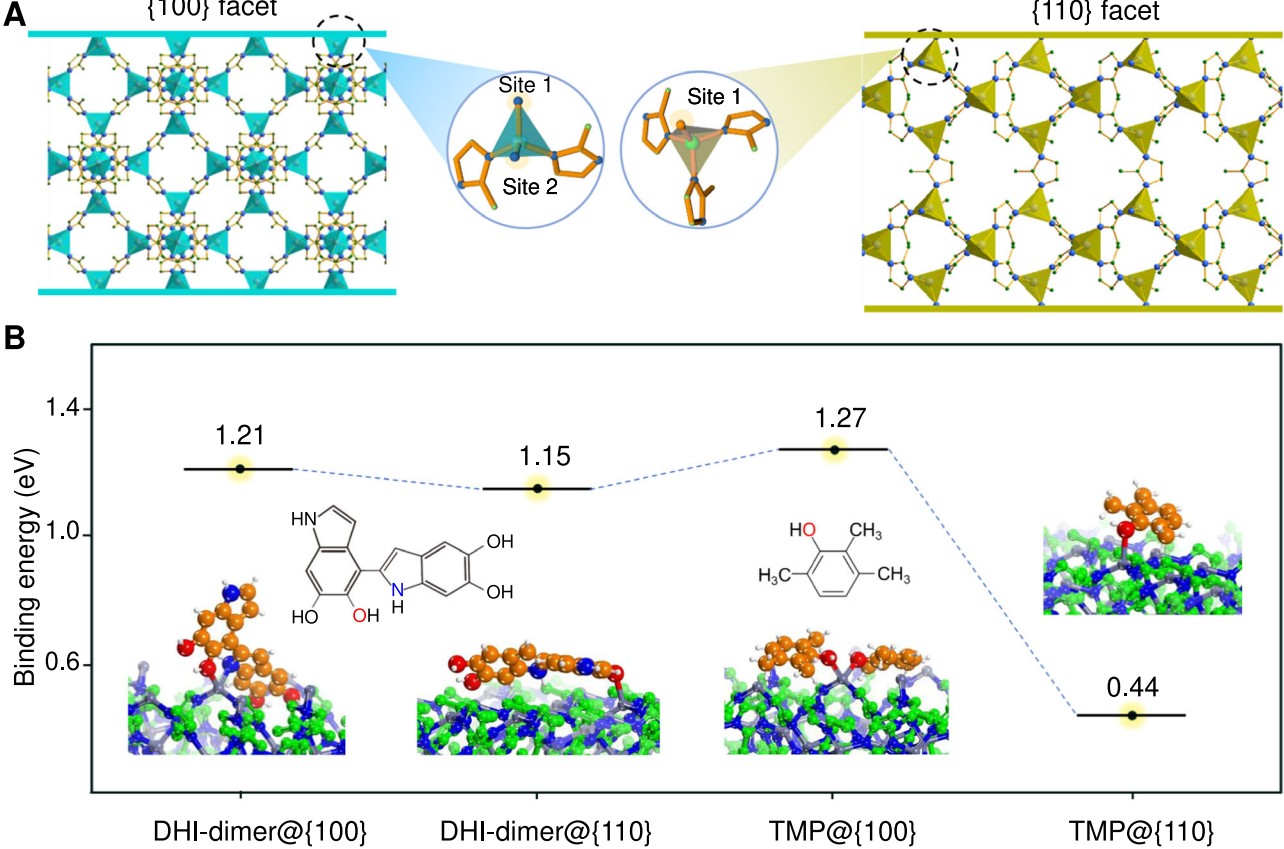

**Fig. 4 | The mechanism of selective occupation strategy. A** Schematic illustration of the coordination environments of the $Zn^{2+}$ at {100} and {110} facets of ZIF-8. **B** Density functional theory (DFT) simulation results of the binding energy of the dopamine oligomer (DHI-dimer) and TMP occultation molecule on {100} and {110} facets of ZIF-8.

micropores of ZIF-8 subunits and small $CuO_2$ nanoparticles in the mesopores of mPDA subunits (Supplementary Fig. 28). At pH 7.4, the structure of type 1 p-ANHs is well maintained; At pH 6.0, the MOF subunits in type 1 p-ANHs is destroyed, leading to the drug release (Supplementary Fig. 30). Since PDA is a good photothermal agent, type 1 p-ANHs can also exhibit good photothermal effect under NIR 808-nm irradiation (Supplementary Fig. 31). Under the acidity tumor microenvironment, the $CuO_2$ in the mPDA can release $Cu^{2+}$. In addition, due to the acidity-induced destruction of MOF subunits, the DSF was also released. The co-accumulation of DSF and $Cu^{2+}$ in tumor cells results in the rapid formation of high cytotoxic bis(N,N-diethyl dithiocarbamato)·$Cu^{2+}$ complexes[34–36]. Moreover, combined with the intrinsic photothermal effect of type 1 p-ANHs, the anticancer effect of the nanodrug is greatly augmented by the hyperthermia upon NIR irradiation, thus inducing remarkable cell apoptosis (Supplementary Fig. 32).

**The selective occupation mechanism**
The emulsion synthesis system involves three phases of solid, aqueous phase and oil phase, i.e., ZIF-8 host nanoparticles, water/ethanol mixed aqueous phase, TMB/TMP mixed oil phase. Surfactants are also essential. We assume that the site-specific anisotropic assembly of amorphous mPDA on the {100} or {110} facets of crystalline ZIF-8 can be attributed to three reasons.

First, the coordination numbers of $Zn^{2+}$ on the {100} and {110} facets of ZIF-8 are different, which is 2 on {100} and 3 on the {110} facets (Fig. 4A)[37,38]. This is the basis of site-specific anisotropic assembly. Based on the Density Functional Theory (DFT), we calculated the binding energies of dopamine oligomer on the {100} and {110} facets of ZIF-8, respectively. Owing to the complexity and uncertainty of

dopamine cross-linking, we select 5,6-dihydroxyindole (DHI)-dimers as the typical representative of dopamine oligomer[39]. The result shows that DHI-dimers can coordinate with $Zn^{2+}$ ions of {100} facets according to bidentate coordination of O and N to form the most stable structure, and the binding energy is as high as 1.21 eV (Fig. 4B). In contrast, the coordination of DHI-dimers with $Zn^{2+}$ ions of {110} facets can only bind in the way of unidentate coordination, and the binding energy is 1.15 eV, lower than the bidentate coordination on the {100} facets. Therefore, when the water/ethanol ratio of aqueous phase is 6:4 and the content of TMP in oil phase is less than 200 mg, the mPDA can be selectively coordinated and assembled on {100} facets of ZIF-8 host.

The second key factor is the amount of TMP in oil phase. TMP can coordinate with unsaturated $Zn^{2+}$ on ZIF-8 host through phenolic hydroxyl group, and then compete with DHI-dimers to adsorb on {100} facets, resulting in the occultation of {100} facets and the selective assembly of mPDA on {110} facets. This assumption can be proved by comparing the binding energy of TMP molecule and DHI-dimers with $Zn^{2+}$ ions of {100} facets (Fig. 4B). The two-coordinated $Zn^{2+}$ of {100} facets can adsorb 2 TMP molecules, and the total binding energy is 1.27 eV, which is obviously higher than that of DHI-dimer. In contrast, on {110} facets, only one TMP molecule can be coordinated, and the binding energy is much lower (0.44 eV), so it cannot compete with DHI-dimer, leading to the selective growth of mPDA on {110} facets of ZIF-8 host. The prerequisite for this result is that TMP in the oil phase can sufficiently contact with ZIF-8 host during the reaction, which is not only related to the content of TMP in the oil phase (>200 mg), but also related to the third factor we discuss below, i.e., water/ethanol ratio.

The water/ethanol ratio directly affects the stability of the emulsion. When the water/ethanol ratio is 6:4, TMB/TMP oil phase can form

stable nanodroplets (Supplementary Fig. 33). In this case, the oil nanodroplets are well encapsulated in the surfactant micelles and cannot contact with ZIF-8 nanoparticles sufficiently. So, the TMP cannot compete with dopamine oligomers. In contrast, the increase of water/ethanol ratio to 8:2 can induce the decrease of emulsion stability, resulting in phase separation and the formation of large metastable oil droplets. Under stirring condition, TMP in metastable oil droplets can contact with ZIF-8 nanoparticles and coordinate with $Zn^{2+}$, which further results in the aforementioned selective assembly of dopamine oligomers on {110} facets. In order to consider the effect of solvent on binding energy, we have checked the solvation effect using implicit solvent model, where the permittivity of the solvent is varied from 78.4 as pure water to 60 and 40 to approximately mimic the ethanol solution at different concentration (Supplementary Table 2). The result demonstrates that the solvation will enhance the adsorption of both DHI and TMP on {100}, and weaken the adsorption of both molecule on {110}. However, the stability trend remains.

In summary, we have demonstrated a selective occupation strategy to achieve site-specific anisotropic growth of amorphous mesoporous polydopamine (mPDA) on crystalline ZIF-8. Four types of p-ANHs with intricate and unprecedented binary and ternary superstructures were synthesized. The selective occupation strategy described here is simple, reproducible, and holds great potential for the construction of p-ANHs with both amorphous and crystalline building blocks. The obtained various p-ANHs with intricate and unprecedented architectures across the research boundary between crystalline p-ANHs and amorphous p-ANHs. These findings reveal the significance of interface chemistry in the controlled synthesis of structurally well-defined hybrid nanocomposites. Moreover, these nanostructures provide a good platform for the construction of nanocomposites with different multiple functionalities and understanding of the structure-property-function relationships.

## Methods

### Chemicals
Zinc acetate dihydrate (Zn(OAC)$_2$·2H$_2$O, 99.99%) and dopamine hydrochloride (AR, 98.0%,) were purchased from Aladdin Corp. Cetyltrimethylammonium bromide (CTAB, 99%) and triblock poly(ethylene oxide)-b-poly(propylene oxide)-b-poly(ethylene oxide) Pluronic F-127 (Mav = 12,600) were purchased from Sigma-Aldrich. Tris(hydroxymethyl)aminomethane (Tris, ≥99.5%), 2-methylimidazole (1-MeIM, AR, ≥98.0%), zinc nitrate hexahydrate (Zn(NO$_3$)$_2$·6H$_2$O, AR, ≥99.0%), cobalt (II) nitrate hexahydrate (Co(NO$_3$)$_2$·2H$_2$O, ≥ 98.5%,), 1,3,5-trimethylbenzene (TMB, ≥ 99.5%), hydrochloric acid (HCl, 36.0–38.0%), 2,3,6-trimethylphenol (TMP, 98.0%), methanol anhydrous (AR, ≥99.5%), hydrogen peroxide (30 wt.%), ammonium hydroxide solution (28 wt.% NH$_3$ in H$_2$O), polyvinylpyrrolidone K30 (GR), copper(II) chloride dihydrate(AR, ≥99.0%) and anhydrous ethanol (AR, ≥99.7%) were purchased from Sinopharm Chemical Reagent Co., Ltd. Disulfiram (97%) was purchased from Aladdin Reagent Co. Ltd. Fetal bovine serum (FBS), penicillin-streptomycin, trypsin, and RPMI 1640 medium were provided by Gibco Life Technologies Co. Cell Counting Kit-8 (CCK-8), calcein-AM/propidium iodide (PI) and annexin V-FITC Apoptosis Detection Kit were obtained from Beyotime Biotech (China).

### Synthesis of truncated rhombic dodecahedral (TRD) ZIF-8 nanoparticles
TRD ZIF-8 nanoparticles were synthesized as reported previously[33]. In a typical synthesis of the TRD ZIF-8 nanoparticles with the diameter of 260 nm and {110}/{100} exposure ratio of 0.43, Zn(OAC)$_2$·2H$_2$O (300.0 mg) was dissolved in 5.0 mL of deionized (DI) water. In a separate beaker, 2-methylimidazole (1170.0 mg) and CTAB (9.7 mg) were dissolved in 5.0 mL of DI water. The two solutions were mixed together in a 100 mL round bottom flask, and the mixture was stirred for a few seconds. Then, the resulting solution was placed at room

temperature for 2.5 h. The obtained TRD ZIF-8 nanoparticles were washed with DI water for three times and collected by centrifuging at 9000 rpm (centrifugal force, 9690 g) for 10 min. The final products were dispersed in 10.0 mL of ethanol (20 mg/mL). The other ZIF-8 nanoparticles with different sizes and {110}/{100} exposure ratios were synthesized by tunning the reaction time and CTAB/2-methylimidazole ratio.

### Synthesis of cubic ZIF-8 nanoparticles
Cubic ZIF-8 nanoparticles were synthesized as reported previously[37]. Zn(OAC)$_2$·2H$_2$O (145.0 mg) was dissolved in 5.0 mL of deionized (DI) water. In a separate beaker, 2-methylimidazole (2267.3 mg) and CTAB (7.0 mg) were dissolved in 35.0 mL of DI water. The two solutions were mixed together in a 50 mL round bottom flask, and the mixture was stirred for another 5 min at 500 rpm. Then, the resulting solution was placed at room temperature for 3.0 hours. The obtained cubic ZIF-8 nanoparticles were washed with DI water for three times and collected by centrifuging at 8000 rpm (centrifugal force, 17726 g) for 10 min.

### Synthesis of type 1 p-ANHs
The type 1 p-ANHs were synthesized through the selective occupation strategy. In a typical process, 20.0 mg of TRD ZIF-8 nanoparticles were dispersed in 4.0 mL of ethanol, 6.0 mL of H$_2$O. Then, 200.0 mg of F-127 and 200.0 mg of dopamine hydrochloride were added and stirred to form a white solution. After that, 1.0 mL of TMB was added. The mixture was stirred for 30 min to form a stable emulsion, and then 0.5 mL of Tris(hydroxymethyl) aminomethane (Tris) aqueous solution (0.15 mM) was added. The reaction solution was left to react at room temperature for 8 hours. Then, the type 1 p-ANHs were centrifuged and washed with ethanol for three times. The final products were dispersed in 5.0 mL of ethanol (5 mg/mL). The mesopore sizes of the mPDA subunits of type 1 p-ANHs can be tuned by introducing TMP in the TMB oil phase. For example, in the typical synthesis without the addition of TMP, the mesopore size is about 5 nm, which can be further increased to 10 and 20 nm as the increase of TMP in the oil phase to 150 and 200 mg, respectively. It worth noting that the amount of TMP must be less than 200 mg.

### Synthesis of type 2 p-ANHs
The type 2 p-ANHs were synthesized through the selective occupation strategy. In a typical process, 20.0 mg of TRD ZIF-8 nanoparticles were dispersed in 2.0 mL of ethanol, 8.0 mL of H$_2$O. Then, 200.0 mg of F-127 and 200.0 mg of dopamine hydrochloride were added and stirred to form a white solution. After that, 1.0 mL of TMB with 250 mg of TMP was added, the mixture was stirred for 30 min to form a milky white emulsion, and then 0.5 mL of Tris aqueous solution (0.15 mM) was added. The reaction solution was left to react at room temperature for 8 hours. Then, the type 2 p-ANHs were centrifuged and washed three times with ethanol.

### Synthesis of type 3 p-ANHs
For the epitaxial growth of ZIF-67 tertiary building blocks on the type 1 p-ANHs, 1.0 mL of type 1 p-ANHs stock solution (4 mg/mL), 3.0 mL of 2-MeIM solution in methanol (25 mM) and 3.0 mL of Co(NO$_3$)$_2$·6H$_2$O solution in methanol (25 mM) were mixed and allowed to react at room temperature for 12 h. The final product was collected by centrifugation, washed with methanol for three times, and dispersed in 5.0 mL methanol for further use. For the epitaxial growth of ZIF-8 tertiary building blocks, the Co(NO$_3$)$_2$·6H$_2$O solution were replaced by Zn(NO$_3$)$_2$·6H$_2$O solution (12.5 mM).

### Synthesis of type 4 p-ANHs
For the epitaxial growth of ZIF-67 tertiary building blocks on the type 1 p-ANHs, 2.0 mL of type 2 p-ANHs stock solution (4 mg/mL), 3.0 mL of 2-MeIM solution in methanol (25 mM) and 3.0 mL of Co(NO$_3$)$_2$·6H$_2$O

solution in methanol (25 mM) were mixed and allowed to react at room temperature for 12 h. The final product was collected by centrifugation, washed with methanol for three times, and dispersed in 5.0 mL methanol for further use. For the epitaxial growth of ZIF-8 tertiary building blocks, the $Co(NO_3)_2 \cdot 6H_2O$ solution were replaced by $Zn(NO_3)_2 \cdot 6H_2O$ solution (12.5 mM).

## Details of density functional theory (DFT) calculation

The (100) surface of ZIF-8 is modeled by a $(1 \times 1)$ slab $(16.92 \times 16.92$ Å), containing 278 atoms in total (17.44 Å thick) with the bottom 140 atoms fixed. The (110) surface is modeled by a $(1 \times 1)$ slab $(16.92 \times 23.93$ Å), containing 508 atoms in total (21.04 Å thick) with the bottom 260 atoms fixed. All DFT calculations were performed by using the plane-wave VASP (Vienna Ab-initio Simulation Package) code[40], where the electron-ion interaction is represented by the projector augmented wave (PAW) potential. The exchange-correlation functional utilized to train NN potential was the GGA-PBE[41], and the kinetic energy cutoff was 450 eV. The first Brillouin zone k-point sampling adopted the Monkhorst-Pack scheme with only the Γ point for large bulk and slab model of {100} and {110} facets. The energy and force criteria for convergence of the electron density and structure optimization were set at $5 \times 10^{-6}$ eV and 0.05 eV/Å, respectively. The dispersion effect is considered by using Grimme's D3 correction (PBE-D3)[42]. An implicit solvation model[43] that developed by Mathew et al. is adopted to further exam the possible solvation effect on the adsorption energy.

## Characterizations

Transmission electron microscopy (TEM), high-resolution transmission electron microscopy (HRTEM), and high-angle annular dark-field images in the scanning TEM (HAADF-STEM) measurements were taken on JEM-2100F microscope (JEOL, Japan) operated at 200 kV. Field-emission scanning electron microscopy (FESEM) images were conducted on a Hitachi Model S-4800 and Gemini Ultra55 (Zeiss). Samples for TEM and FESEM tests were prepared by drying the nanoparticles on amorphous carbon-coated copper grids. Tridium). Nitrogen adsorption-desorption measurements were conducted to obtain information on the porosity. The measurements were conducted at 77 K with ASAP 2420 and Micromeritcs Tristar 3020 analyzer (USA). Dynamic light scattering (DLS) analysis was conducted on a Malvern ZS90 with a He, Ne laser (633 nm, 4 mW). Confocal fluorescence images were obtained by an LSM 710 confocal laser scanning microscope (Carl Zeiss SMT Inc., USA). Flow cytometry analysis was performed by an Accuri C6 flow cytometer (BD Biosciences, USA).

## Data availability

All the data generated in this study are provided in the main text and Supplementary Information. Source data are provided with this paper.

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

## Acknowledgements

We gratefully acknowledge funding from the National Natural Science Foundation of China (22075049 (X.L.), 21875043 (X.L.), 22088101 (D.Z.), 21701027 (X.L.), 21733003 (D.Z.), 51961145403 (F.Z.)), National Key R&D Program of China (2018YFA0209401) (D.Z.), Fundamental Research Funds for the Central Universities (20720220010) (X.L.), Key Basic Research Program of Science and Technology Commission of Shanghai Municipality (17JC1400100 (F.Z.), 22JC1410200 (D.Z.)), Natural Science Foundation of Shanghai (22ZR1478900 (X.L.), 18ZR1404600 (X.L.), 20490710600 (X.L.)), Shanghai Rising-Star Program (20QA1401200 (X.L.), 22YF1402200 (T.Z.)), Qatar National Research Fund (a member of the Qatar Foundation) (NPRP Grant # NPRP 12S-0309-190268 (X.L.))

## Author contributions

M.L. and X.L. conceived the project. M.L. synthesized and characterized the materials, and performed SEM and TEM characterizations. C.S. performed DFT. Z.L. and Y.K. conducted structural modeling and PXRD refinements. M.L., C.S., T.Z., H.Y., Y.K., Z.L., M.H., F.Z., Q.L., D.Z., and X.L. analyzed the data and discussed the experimental work. All authors discussed the results and contributed to the manuscript.

## Competing interests

The authors declare no competing interests.
