## [Peer Review File · Nature Communications]

Site-specific anisotropic assembly of amorphous mesoporous subunits on crystalline metal-organic frameworkREVIEWER COMMENTS

Reviewer #1 (Remarks to the Author):

This work reports an interesting method to anisotropically grow mPDA on different facets of ZIF-8 to form the p-ANHs. The manuscript was well written and the synthetic mechanism was also convincing. However, the applications of the p-ANHs were missing despite the authors implied many prospects in the introduction such as guest delivery, nanomedicine, catalysis and so on. Besides, the generality of the approach was not demonstrated. Could the anisotropic growth of mPDA be applied on other kinds of MOFs? Or could the other amorphous mesoporous subunits be anisotropically growth on ZIF-8? The manuscript can be further improved with a few more experiments. Here are some specific comments.

- I understand this might be the first example of amorphous p-ANHs on crystalline substrate combination. I am curious about the differences for their potential applications. What are the different application advantages of crystalline p-ANHs on crystalline substrate, amorphous p-ANHs on amorphous substrate and amorphous p-ANHs on crystalline substrate in this case?
- In the main manuscript on page 5, TMP can be used to tune the mesopore sizes of mPDA. What are the possible explanations for the mechanism? Is the mesopore size of type 2 mPDA still tunable if >200 mg TMP is minimum required for type 2 growth.
- In Supplementary Figure. 10, cubic ZIF-8 containing 100% {100} facet was used to grow type 2 mPDA. The result showed mPDA grown on the edge of cubic ZIF-8. I am curious what are the results if using RD ZIF-8 which is 100% {110} facet for type 2 growth. In addition, do the contrary results happen if growing type 1 mPDA on cubic ZIF-8 and RD ZIF-8?
- In Type 3 & 4 ternary-superstructures, the authors described that mPDA nanoplates inhibit the deposition of ZIF-67 so ZIF-67 can epitaxially grow on the exposed facets of pristine ZIF-8 host. Do the authors see any evidences to exclude the possibility that ZIF-67 could also grow in mesopore area of mPDA? The pore area of mPDA contained exposed ZIF-8 as well.
- The experimental detail of DFT calculation is missing. Did the authors consider solvent effects of binding energy calculation since the solvent conditions of type 1 and type 2 were very different? In addition, the difference of binding energies between TMP on the {100} and {110} facets of ZIF-8 are very large due to coordination number difference. I was wondering why binding energies of dopamine oligomer on the {100} and {110} facets of ZIF-8 are so closed. The binding energies of dopamine oligomer should also highly relate to the coordination numbers of different facets of ZIF-8.

Reviewer #2 (Remarks to the Author):

Nano composite of MOF nano particles and mesoporous polydopamine were prepared by the facet-selective surface modification approach. The resultant hybrid composite showed unique morphology which could be controlled by the synthetic conditions of solution processing. Microscope images of the composites are the beautiful piece of work. Actually, I enjoyed reading the manuscript very much. However, the approach is well-established as far as I understood. The facet-selective modification by functional molecules have been widely used to control the shape of nanoparticles, growth of designated crystals on the selected facet of the single crystals, preparation of Janus nanoparticles, and others. (for example, 10.1021/acs.jpcc.1c06408) Although the obtained composite particle are beautiful, the manuscript lacking the novelty in the sense of the chemistry. In addition, the novel or enhanced functionalities by the composite formation are required to publish the manuscript in the sophisticated journals like Nature Communications. Accordingly, the manuscript is published in the more specified journal related to the chemistry or materials science.

Reviewer #3 (Remarks to the Author):

In this manuscript titled as "Site-specific anisotropic assembly of amorphous mesoporous subunits on crystalline metal-organic framework", the authors demonstrated a nice example of the anisotropic growth of amorphous mesoporous mPDA on ZIF-8 with a selective occupation strategy. It is interesting that four types of p-ANHs including the mPDA nanoplates and hybrid

superstructures were synthesized in a controllable way. This research was well organized and the results were presented in a very nice way. However, there are some problems that making it not good enough to be published in Nature Communications in current form and a major revision is needed.

1. In the synthesis of type 1 and type 2 p-ANHs, ZIF-8 was dispersed in the EtOH/H₂O mixtures, "Then, 200.0 mg of F-127 and 200.0 mg of dopamine hydrochloride were added and stirred to form a clear solution", were the ZIF-8 nanoparticles dissolved in this process?
2. The calculated BET surface area of ZIF-8 in p-ANHs is 1086 m²g⁻¹, the authors thought it was "nearly to that of the pristine ZIF-8 nanoparticles (Table S1)". However, the BET surface area of pristine ZIF-8 was 1457 m²g⁻¹ (see Table S1). There is a 27% large difference and therefore the authors should give a reasonable explanation.
3. ZIF-8 has excellent chemical stability but is prone to degrade under acidic conditions. The {100} facet and {110} facet were degraded to different degrees as the dopamine hydrochloride was added. It seems that Zn was detected in the type 1 p-ANHs. Is It possible the defects formed and induced the anisotropic growth of mPDA?
4. For ZIF-8, the {110} facet contains six-membered rings in the surface plane. Please check Figure 4A on page 14.
5. On page4 "the TRD ZIF-8 nanoparticles are used as a host for the deposition of dopamine precursors under alkaline condition", the pH value should be indicated.
6. In the Methods section, the Full name of Tris should be marked before the abbreviation used in "Tris aqueous solution".
7. In conclusion section, the authors emphasized they "achieve f site-specific anisotropic growth", what was the "f site"? they did not mentioned it in main text of the manuscript. Besides, "The research "boundary" between the crystalline p-ANHs and amorphous p-ANHs is crossed" and "The obtained various p-ANHs with intricate and unprecedented architectures across the research boundary between crystalline p-ANHs and amorphous p-ANHs", were the two sentences repeated?

Point-by-Point Response to Referees

Reviewer #1:

This work reports an interesting method to anisotropically grow mPDA on different facets of ZIF-8 to form the p-ANHs. The manuscript was well written and the synthetic mechanism was also convincing. However, the applications of the p-ANHs were missing despite the authors implied many prospects in the introduction such as guest delivery, nanomedicine, catalysis and so on. Besides, the generality of the approach was not demonstrated. Could the anisotropic growth of mPDA be applied on other kinds of MOFs? Or could the other amorphous mesoporous subunits be anisotropically growth on ZIF-8? The manuscript can be further improved with a few more experiments. Here are some specific comments.

Answer: We greatly appreciate the reviewer's useful comments and suggestions. We have carefully revised the manuscript based on the comments. In the revised manuscript, the applications of the obtained p-ANHs and the generality of the synthesis method were further verified. For the applications of the obtained p-ANHs, the multiple-drugs co-delivery, photothermal performance, and cancer therapy of the obtained type 1 p-ANHs were studied. For the generality of the synthesis, we found that the anisotropic growth of mPDA can also be applied to titanium MOF of MIL-125, and the mSiO₂ can also anisotropically grow on the {100} facets of ZIF-8.

Correspondingly, we have added the following description on Page 8 of the revised manuscript:

The abundant micropores in the MOF subunits and mesopores in the mPDA subunits make the p-ANHs a good nanocarriers for multiple drugs co-delivery, and the excellent photothermal performance of mPDA subunits upon 808-nm near infrared (NIR) laser irradiation further endows p-ANHs with great potential as effective multifunctional therapeutic agent (Supplementary Fig. 29, Fig. 31 and Fig. 32). As a proof of concept, the loading of cargoes, pH responsive drug release, photothermal performance and anti-tumor ability of the obtained type1 p-ANHs were investigated. Type 1 p-ANHs were used as a model for the co-loading of disulfiram (DSF) in micropores of ZIF-8 subunits and small CuO₂ nanoparticles in the mesopores of mPDA subunits (Supplementary Fig. 28). At pH 7.4, the structure of type 1 p-ANHs is well maintained; At pH 6.0, the MOF subunits in type 1 p-ANHs is destroyed, leading to the drug release (Supplementary Fig. 30). Since PDA is a good photothermal agent, type 1 p-ANHs can also exhibit good photothermal effect under NIR 808-nm irradiation (Supplementary Fig. 31). Under the acidity tumor microenvironment, the CuO₂ in the mPDA can release Cu²⁺. In addition, due to the acidity-induced destruction of MOF subunits, the DSF was also released. The co-accumulation of DSF and Cu²⁺ in tumor cells results in the rapid formation of high cytotoxic bis(N,N-diethyl dithiocarbamate)-Cu²⁺ complexes³⁴⁻³⁶. Moreover, combined with the intrinsic photothermal effect of type 1 p-ANHs, the anticancer effect of the nanodrug is greatly augmented by the hyperthermia upon NIR irradiation, thus inducing remarkable cell apoptosis (Supplementary Fig. 32).

We have added the following description on Page 6 of the revised manuscript: Moreover, the anisotropic growth of mPDA can also be applied to titanium MOF of MIL-125, and amorphous mSiO₂ nanoplates can also be anisotropically grown on the {100} facets of TRD ZIF-8 (Supplementary Fig. 10 and Fig. 11).

We have added the following Supplementary Figures in the revised Supporting Information:

Supplementary Figure 28. (A) TEM image and (B) dynamic light scattering (DLS) measurement of PVP-coated CuO_2 nanodots. It can be seen that the obtained CuO_2 nanodots have a small size of ~ 7 nm. The CuO_2 nanodots could be well-dispersed in water with a mean hydrodynamic diameter of ~ 17.3 nm. Scale bars: 100 nm.

Supplementary Figure 29. (A) Schematic illustration of the synthetic process of CuO_2 &DSF-loaded type 1 p-ANHs. (B) TEM image, (C) dark-field TEM image and corresponding elemental mappings of CuO_2 &DSF-loaded type 1 p-ANHs. (D–F) XPS spectra of CuO_2 &DSF-loaded type 1 p-ANHs in different binding-energy ranges. Scale bars: 200 nm.

Supplementary Figure 30. TEM images of type 1 p-ANHs after 12 h storage in PBS buffers at (A) pH 7.4 and (B) 6.0. Scale bars: 200 nm. It can be seen that the morphology and structure of type 1 p-ANHs remain intact under normal physiological conditions of pH 7.4, while type 1 p-ANHs decomposes to lamellar dopamine nanosheets under weak acidic condition. These results indicate that type 1 p-ANHs has good pH responsiveness.

Supplementary Figure 31. (A) Heating curves of CuO₂&DSF-loaded type 1 p-ANHs with different concentrations upon 808-nm laser irradiation (1.25 W/cm²). (B) Heating curves of CuO₂&DSF-loaded type 1 p-ANHs aqueous dispersion (300 μg/mL) upon 808-nm laser irradiation at varying power densities. (C) Infrared (IR) thermographic images of CuO₂&DSF-loaded type 1 p-ANHs dispersions with different concentrations under 808-nm laser irradiation (1.25 W/cm²). (D) Heating curve of a CuO₂&DSF-loaded type 1 p-ANHs aqueous dispersion (300 μg/mL) under four cycles of heating and cooling processes (1.25 W/cm²). (E) Photo-to-heat conversion capability of a CuO₂&DSF-loaded type 1 p-ANHs aqueous dispersion (300 μg/mL) upon 808-nm laser irradiation (1.25 W/cm²). (F) IR thermographic images of CuO₂&DSF-loaded type 1 p-ANHs solution (300 μg/mL) under the irradiation of 808-nm laser with varied power densities.

Supplementary Figure 32. (A–C) Viability of HeLa cells after different treatments for 24 h. (D) CLSM images of HeLa cells after incubation with PBS, free DSF, CuO_2 , Laser, CuO_2 &DSF-loaded type 1 p-ANHs, and CuO_2 &DSF-loaded type 1 p-ANHs + Laser for 24 h. (E) Quantitative analysis of HeLa cells by flow cytometry, including the percentages of live, early apoptotic, and late apoptotic cells after different treatments. Scale bars: 100 μm . It can be seen that type 1 p-ANHs, CuO_2 and DSF alone show negligible toxicity to HeLa cells, even at high concentrations. In comparison, more than 40% of cells were killed after the CuO_2 &DSF-loaded type 1 p-ANHs treatment (10 $\mu\text{g/mL}$). Furthermore, the chemotherapeutic efficacy of CuO_2 &DSF-loaded type 1 p-ANHs was significantly increased when the cells were further treated with NIR laser.

Supplementary Figure 10. (A) SEM image of MIL-125 based p-ANHs synthesized by the selective growth of mPDA on {001} facets of MIL-125. (B, C) SEM images of a single MIL-125 p-ANHs taken from different perspectives. Scale bars: 500 nm.

Supplementary Figure 11. SEM image of mSiO₂&ZIF-8 p-ANHs. It can be seen that amorphous mSiO₂ is selectively grown on {100} facets of TRD ZIF-8. Scale bars: 200 nm.

The corresponding materials, methods and other relevant contents have also been added in the revised Supporting Information.

Comments 1. *I understand this might be the first example of amorphous p-ANHs on crystalline substrate combination. I am curious about the differences for their potential applications. What are the different application advantages of crystalline p-ANHs on crystalline substrate, amorphous p-ANHs on amorphous substrate and amorphous p-ANHs on crystalline substrate in this case?*

Answer: Thanks very much for the useful comments. Compared to crystalline-crystalline (microporous) and amorphous-amorphous (mesoporous) p-ANHs, the feature of amorphous-crystalline p-ANHs reported in this manuscript is the ability to provide two different sets of pores, *i.e.* micropores in the crystalline MOF subunits and the mesopores in the mPDA subunits. So, the obtained amorphous-crystalline p-ANHs can be used for the co-loading of multiple cargoes with different sizes. In comparison, crystalline-crystalline p-ANHs generally have only microporous structure, and amorphous-amorphous p-ANHs generally have only mesoporous structure. So, they are difficult to realize the co-loading of multiple cargoes with quite different sizes. In the revised manuscript, the obtained type 1 p-ANHs were used as a model for the co-loading of small molecule disulfiram (DSF) in micropores of ZIF-8 subunits and small CuO₂ nanoparticles in the mesopores of mPDA subunits, and the obtained CuO₂&DSF-loaded type 1 p-ANHs were used for killing cancer cells. The relevant results have been presented in the above contents and revised manuscript.

Comments 2. *In the main manuscript on page 5, TMP can be used to tune the mesopore sizes of mPDA. What are the possible explanations for the mechanism? Is the mesopore size of type 2 mPDA still tunable if >200 mg TMP is minimum required for type 2 growth.*

Answer: We thank the reviewer very much for the questions. This synthesis method is an emulsion system formed by TMB/TMP mixed oil phase, water/ethanol mixed aqueous phase and F127 surfactants. TMB/TMP molecules can interact with the hydrophobic PPO segment of F127 molecules through van der Waals forces. So, the F127 micelles can be swollen by TMB/TMP oil phase, which further result in the size increase of micelles and mesopores. It is worth mentioning

that TMP in the oil phase may further enhance the swelling effect of oil on micelles through strong hydrogen bonding with the PPO segment. So, the mesopore size of type-1 architecture can be well tuned from 5 to 20 nm by tuning the TMP from 0 to 200 mg. Similarly, the mesopore size of type-2 architecture can be well tuned from 20 to 80 nm when the TMP in the oil phase was increased from 200 to 300 mg.

Correspondingly, we have added the following description on page 6 of the revised manuscript: Moreover, the pore size of mPDA could be increased from 20 to 80 nm when the TMP in the oil phase was increased from 200 to 300 mg (Supplementary Fig. 13).

We have also added Supplementary Figure 13 in the revised Supporting Information:

Supplementary Figure 13. TEM images of type 2 p-ANHs with tunable mesopore sizes. Scale bars: 100 nm. The mesopore size of mPDA subunits can be tuned by varying the ratio of TMP/TMB in the oil phase. As the amount of TMP in the oil phase increases from 200 to 300 mg, the pore size can be adjusted from 20 to 80 nm. It worth noting that the amount of TMP must be more than 200 mg for the synthesis of type 2 p-ANHs.

Comments 3. *In Supplementary Figure. 10, cubic ZIF-8 containing 100% {100} facet was used to grow type 2 mPDA. The result showed mPDA grown on the edge of cubic ZIF-8. I am curious what are the results if using RD ZIF-8 which is 100% {110} facet for type 2 growth. In addition, do the contrary results happen if growing type 1 mPDA on cubic ZIF-8 and RD ZIF-8?*

Answer: We thank the reviewer very much for the insightful comments. We have accepted them. For type 2 growth, if RD ZIF-8 is used as host, mPDA subunits grow completely on the {110} facets of RD ZIF-8 to form the core@shell structure. For type 1 growth, if RD ZIF-8 is used as host, the growth manner of mPDA is changed greatly. Instead of the layer growth to form the mPDA nanoplates, many mPDA nanoparticles with varying sizes are deposited on the {110} facets of RD ZIF-8. If cubic ZIF-8 is used as host for type 1 growth, mPDA subunits grow completely on the

{100} facets of the nanocube to form the core@shell structure. All the relevant results have been added in the revised manuscript.

Correspondingly, we have added the following description on page 6 of the revised manuscript:

In addition, by using the TRD ZIF-8 with different {110}/{100} exposure ratio as pristine host, the selective growth of mPDA subunits can still be carried out (Supplementary Fig. 8 and Fig. 9).

We have added the following description on page 8 of the revised manuscript: The width of the mPDA nanoplate can be varied by tuning the {100}/{110} exposure ratio on the pristine TRD ZIF-8 host (Supplementary Fig. 14).

We have also added Supplementary Figure 9 and Supplementary Figure 14 in the revised Supporting Information:

Supplementary Figure 9. The selective growth of mPDA nanoplates on the {100} facets of pristine TRD ZIF-8 host with different {100}/{110} exposure ratio: (A) ~0%; (B) ~100%. Scale bars: 500 nm for (A); 100 nm for (B). For type 1 growth, if RD ZIF-8 is used as host, the growth manner of mPDA is changed greatly. Instead of the layer growth to form the mPDA nanoplates, many mPDA nanoparticles varying sizes are deposited on the {110} facet of RD ZIF-8. If cubic ZIF-8 is used as host for the type 1 growth, mPDA nanoplates grow completely on the {100} facets of the nanocube to form the core@shell structure.

Supplementary Figure 14. The selective growth of mPDA nanoplates on the $\{110\}$ facets of pristine TRD ZIF-8 host with different $\{110\}/\{100\}$ exposure ratio: (A) $\sim 0\%$; (B) $\sim 43\%$. (C) $\sim 100\%$, Scale bars: 200 nm for (A); 100 nm for (B); 500 nm for (C). When RD ZIF-8 with 100% $\{110\}$ facets exposure is used as host, mPDA subunits grow completely on the $\{110\}$ facets of RD ZIF-8 to form the core@shell structure. Regardless of $\{110\}/\{100\}$ exposure ratio, even close to 0%, mPDA nanoplates can selectively grow on $\{110\}$ planes of ZIF-8 host, indicating the universality of the selective occupation strategy.

Comments 4. *In Type 3 & 4 ternary-superstructures, the authors described that mPDA nanoplates inhibit the deposition of ZIF-67 so ZIF-67 can epitaxially grow on the exposed facets of pristine ZIF-8 host. Do the authors see any evidences to exclude the possibility that ZIF-67 could also grow in mesopore area of mPDA? The pore area of mPDA contained exposed ZIF-8*

as well.

Answer: Thanks very much for the comment. In the revised manuscript, we have added more results about the deposition of ZIF-67 in the mesopore channels of mPDA subunits. The results show that the growth of ZIF-67 in mesopore channels of mPDA subunits depends on the volume of added Co^{2+} and 2-Methylimidazole (2-MeIM) solutions. For example, when 3.0 mL of Co^{2+} solution (12.5 mM) and 3.0 mL of 2-MeIM solution (25 mM) were added, ZIF-67 could only grow epitaxially along the exposed $\{110\}$ facets of pristine ZIF-8. When 5.0 mL of Co^{2+} solution (12.5 mM) and 5.0 mL of 2-MeIM solution (25 mM) were added, ZIF-67 could not only grow epitaxially along the $\{110\}$ surface of the pristine ZIF-8, but also in small amounts within the mesopore channels of the mPDA subunits. When Co^{2+} and 2-MeIM solutions were further increased to 7.0 mL, although the epitaxial growth manner of ZIF-67 along the exposed $\{110\}$ facets of ZIF-8 was still well maintained, there are so many growth sites can be clearly observed in the mesopore channels of mPDA subunits.

Correspondingly, we have added the following description on page 8 of the revised manuscript:

In addition, it worth noting that the growth of ZIF-67 in mesopore channels of mPDA subunits depends on the volume of added Co^{2+} and 2-Methylimidazole (2-MeIM) solutions (Supplementary Fig. 26 and 27).

We have also added Supplementary Figure 26 and Supplementary Figure 27 in the revised Supporting Information:

Supplementary Figure 26. TEM images of type 3 p-ANHs synthesized by using different amount of $\text{Co}(\text{NO}_3)_2$ solution and 2-MeIM solution. (A) 3.0 mL of 2-MeIM solution (25 mM) and 3.0 mL

of $\text{Co}(\text{NO}_3)_2$ solution (12.5 mM); (B) 5.0 mL of 2-MeIM solution (25 mM) and 5.0 mL of $\text{Co}(\text{NO}_3)_2$ solution (12.5 mM); (C) 7.0 mL of 2-MeIM solution (25 mM) and 7.0 mL of $\text{Co}(\text{NO}_3)_2$ solution. Scale bars: 100 nm.

Supplementary Figure 27. TEM images of type 4 p-ANs synthesized by using different amount of $\text{Co}(\text{NO}_3)_2$ solution and 2-MeIM solution. (A) 3.0 mL of 2-MeIM solution (25 mM) and 3.0 mL of $\text{Co}(\text{NO}_3)_2$ solution (12.5 mM); (B) 5.0 mL of 2-MeIM solution (25 mM) and 5.0 mL of $\text{Co}(\text{NO}_3)_2$ solution (12.5 mM). Scale bars: 200 nm.

Comments 5. *The experimental detail of DFT calculation is missing. Did the authors consider solvent effects of binding energy calculation since the solvent conditions of type 1 and type 2 were very different? In addition, the difference of binding energies between TMP on the {100} and {110} facets of ZIF-8 are very large due to coordination number difference. I was wondering why binding energies of dopamine oligomer on the {100} and {110} facets of ZIF-8 are so closed. The binding energies of dopamine oligomer should also highly relate to the coordination numbers of different facets of ZIF-8.*

Answer: Thank you very much for the constructive question. We have accepted it. In the revised manuscript, we have added the details of DFT calculation in the supplementary information. The model was originally computed in vacuum. Taking referee's advice, we have checked the solvation effect using implicit solvent model, where the permittivity of the solvent is varied from 78.4 as pure water to 60 and 40 to approximately mimic the ethanol solution at different concentration. The result demonstrates that the solvation will enhance the adsorption of both DHI and TMP on {100}, and weaken the adsorption of both molecule on {110}. However, the stability trend remains. The manuscript and the supplementary information have been modified correspondingly. The adsorption energy is not only affected by the coordinate number, but also the Van de Waals interaction between the adsorbate and the surface. The closer contact of DHI molecule to the {110} facets than that on

{100} have also contribute a lot to the adsorption energy, resulting in the smaller energy differences of DHI on different surfaces than that of TMP.

Correspondingly, we have added the following description on page 10 of the revised manuscript:

In order to consider the effect of solvent on binding energy, we have checked the solvation effect using implicit solvent model, where the permittivity of the solvent is varied from 78.4 as pure water to 60 and 40 to approximately mimic the ethanol solution at different concentration. The result demonstrates that the solvation will enhance the adsorption of both DHI and TMP on {100}, and weaken the adsorption of both molecule on {110}. However, the stability trend remains.

We have also added details of the DFT calculation and Table S2 in the revised Supporting Information:

The (100) surface of ZIF-8 is modeled by a (1×1) slab (16.92 Å× 16.92 Å), containing 278 atoms in total (17.44 Å thick) with the bottom 140 atoms fixed. The {110} facets is modeled by a (1×1) slab (16.92 Å× 23.93 Å), containing 508 atoms in total (21.04 Å thick) with the bottom 260 atoms fixed. All DFT calculations were performed by using the plane-wave VASP (Vienna Ab-initio Simulation Package) code⁵, where the electron-ion interaction is represented by the projector augmented wave (PAW) potential. The exchange-correlation functional utilized to train NN potential was the GGA-PBE⁶, and the kinetic energy cutoff was 450 eV. The first Brillouin zone k-point sampling adopted the Monkhorst-Pack scheme with only the Γ point for large bulk and slab model of {100} and {110} facets. The energy and force criteria for convergence of the electron density and structure optimization were set at 5×10^{-6} eV and 0.05 eV/Å, respectively. The dispersion effect is considered by using Grimme's D3 correction (PBE-D3)⁷. An implicit solvation model⁸ that developed by Mathew et al. is adopted to further exam the possible solvation effect on the adsorption energy.

Table S2. The adsorption energy of DHI-dimer and TMP on {100} and {110} facets with the presence of implicit solvent model. All the configuration are firstly optimized in vacuum, followed by single point energy calculations with different value of permittivity setup. (unit: eV)

	DHI/{100}	TMP/{100}	DHI/{110}	TMP/{110}
No solvent	1.21	1.15	1.27	0.44
$\epsilon=78.4$	1.77	1.99	0.69	0.18
$\epsilon=60.0$	1.76	1.97	0.72	0.20
$\epsilon=40.0$	1.74	1.94	0.76	0.22

Reviewer #2:

Nano composite of MOF nano particles and mesoporous polydopamine were prepared by the facet-selective surface modification approach. The resultant hybrid composite showed unique morphology which could be controlled by the synthetic conditions of solution processing. Microscope images of the composites are the beautiful piece of work. Actually, I enjoyed reading the manuscript very much. However, the approach is well-established as far as I understood. The facet-selective modification by functional molecules have been widely used to control the shape of nanoparticles, growth of designated crystals on the selected facet of the single crystals, preparation of Janus nanoparticles, and others. (for example, 10.1021/acs.jpcc.1c06408) Although the obtained composite particle are beautiful, the manuscript lacking the novelty in the sense of the chemistry. In addition, the novel or enhanced functionalities by the composite formation are required to publish the manuscript in the sophisticated journals like Nature Communications. Accordingly, the manuscript is published in the more specified journal related to the chemistry or materials science.

Answer: We thank the reviewer very much for the positive comments. Our mechanism does have similarity with other works in terms of selective adsorption of functional molecules on different crystalline facets. However, different from the morphology control of the nanoparticles, growth of designated crystals on the selected facet of the single crystals, this work focuses on the controllable synthesis of porous anisotropic nanohybrids (p-ANHs). This is the first report on the site-specific anisotropic assembly of amorphous mesoporous subunits on crystalline metal-organic framework. In addition, our synthetic system is a two-phase system of oil and water, in which the micelles were self-assembly on the crystalline facets to form mPDA subunits. So, the synthesis method is different from the reported epitaxial growth method. Furthermore, the amorphous mesoporous subunits (mPDA, mSiO₂) have no lattice matching requirement for the selective growth on the crystalline facets, while the reported epitaxial growth of crystalline facets has a requirement for lattice matching. For the enhanced functionalities (applications) of the obtained p-ANHs, compared to crystalline-crystalline (microporous) and amorphous-amorphous (mesoporous) p-ANHs, the feature of amorphous-crystalline p-ANHs reported in this manuscript is the ability to provide two different sets of pores, *i.e.* micropores in the crystalline MOF subunits and the mesopores in the mPDA subunits. So, the obtained amorphous-crystalline p-ANHs can be used for the co-loading of multiple cargoes with different sizes. In comparison, crystalline-crystalline p-ANHs generally have only microporous structure and amorphous-amorphous p-ANHs generally have only mesoporous structure, so it is difficult to realize the co-loading of multiple cargoes with quite different sizes. In the revised manuscript, the obtained type 1 p-ANHs were used as a model for the co-loading of small molecule disulfiram (DSF) in micropores of ZIF-8 subunits and small CuO₂ nanoparticles in the mesopores of mPDA subunits, and the obtained CuO₂&DSF-loaded type 1 p-ANHs were used for killing cancer cells.

In addition, the generality of the synthesis was further verified in the revised manuscript, we found that the anisotropic growth of mPDA can also be applied to titanium MOF of MIL-125, and the mSiO₂ can also anisotropically grown on the {100} facets of ZIF-8.

All the relevant results have been provided in the revised manuscript.

Correspondingly, we have added the following description on Page 8 of the revised manuscript:

Supplementary Figure 28. (A) TEM image and (B) dynamic light scattering (DLS) measurement of PVP-coated CuO_2 nanodots. It can be seen that the obtained CuO_2 nanodots have a small size of ~ 7 nm. The CuO_2 nanodots could be well-dispersed in water with a mean hydrodynamic diameter of ~ 17.3 nm. Scale bars: 100 nm.

Supplementary Figure 29. (A) Schematic illustration of the synthetic process of CuO_2 &DSF-loaded type 1 p-ANHs. (B) TEM image, (C) dark-field TEM image and corresponding elemental mappings of CuO_2 &DSF-loaded type 1 p-ANHs. (D–F) XPS spectra of CuO_2 &DSF-loaded type 1 p-ANHs in different binding-energy ranges. Scale bars: 200 nm.

Supplementary Figure 30. TEM images of type 1 p-ANHs after 12 h storage in PBS buffers at (A) pH 7.4 and (B) 6.0. Scale bars: 200 nm. It can be seen that the morphology and structure of type 1 p-ANHs remain intact under normal physiological conditions of pH 7.4, while type 1 p-ANHs decomposes to lamellar dopamine nanosheets under weak acidic condition. These results indicate that type 1 p-ANHs has good pH responsiveness.

Supplementary Figure 31. (A) Heating curves of CuO₂&DSF-loaded type 1 p-ANHs with different concentrations upon 808-nm laser irradiation (1.25 W/cm²). (B) Heating curves of CuO₂&DSF-loaded type 1 p-ANHs aqueous dispersion (300 μg/mL) upon 808-nm laser irradiation at varying power densities. (C) Infrared (IR) thermographic images of CuO₂&DSF-loaded type 1 p-ANHs dispersions with different concentrations under 808-nm laser irradiation (1.25 W/cm²). (D) Heating curve of a CuO₂&DSF-loaded type 1 p-ANHs aqueous dispersion (300 μg/mL) under four cycles of heating and cooling processes (1.25 W/cm²). (E) Photo-to-heat conversion capability of a CuO₂&DSF-loaded type 1 p-ANHs aqueous dispersion (300 μg/mL) upon 808-nm laser irradiation (1.25 W/cm²). (F) IR thermographic images of CuO₂&DSF-loaded type 1 p-ANHs solution (300 μg/mL) under the irradiation of 808-nm laser with varied power densities.

Supplementary Figure 32. (A–C) Viability of HeLa cells after different treatments for 24 h. (D) CLSM images of HeLa cells after incubation with PBS, free DSF, CuO_2 , Laser, CuO_2 &DSF-loaded type 1 p-ANHs, and CuO_2 &DSF-loaded type 1 p-ANHs + Laser for 24 h. (E) Quantitative analysis of HeLa cells by flow cytometry, including the percentages of live, early apoptotic, and late apoptotic cells after different treatments. Scale bars: 100 μm . It can be seen that type 1 p-ANHs, CuO_2 and DSF alone show negligible toxicity to HeLa cells, even at high concentrations. In comparison, more than 40% of cells were killed after the CuO_2 &DSF-loaded type 1 p-ANHs treatment (10 $\mu\text{g/mL}$). Furthermore, the chemotherapeutic efficacy of CuO_2 &DSF-loaded type 1 p-ANHs was significantly increased when the cells were further treated with NIR laser.

Supplementary Figure 10. (A) SEM image of MIL-125 based p-ANHs synthesized by the selective growth of mPDA on {001} facets of MIL-125. (B, C) SEM images of a single MIL-125 p-ANHs taken from different perspectives. Scale bars: 500 nm.

Supplementary Figure 11. SEM image of mSiO₂&ZIF-8 p-ANHs. It can be seen that amorphous mSiO₂ is selectively grown on {100} facets of TRD ZIF-8. Scale bars: 200 nm.

The corresponding materials, methods and other relevant contents have also been added in the revised Supporting Information.

Reviewer #3:

In this manuscript titled as “Site-specific anisotropic assembly of amorphous mesoporous subunits on crystalline metal–organic framework”, the authors demonstrated a nice example of the anisotropic growth of amorphous mesoporous mPDA on ZIF-8 with a selective occupation strategy. It is interesting that four types of p-ANHs including the mPDA nanoplates and hybrid superstructures were synthesized in a controllable way. This research was well organized and the results were presented in a very nice way. However, there are some problems that making it not good enough to be published in Nature Communications in current form and a major revision is needed.

Answer: We really appreciate the reviewer’s useful comments and suggestions. We have carefully revised the manuscript based on the suggestions and comments.

Comments 1. *In the synthesis of type 1 and type 2 p-ANHs, ZIF-8 was dispersed in the EtOH/H₂O mixtures, “Then, 200.0 mg of F-127 and 200.0 mg of dopamine hydrochloride were added and stirred to form a clear solution”, were the ZIF-8 nanoparticles dissolved in this process?*

Answer: We thank the reviewer very much. We are sorry for the misunderstanding. The ZIF-8 nanoparticles were not dissolved. Actually, the ZIF-8 nanoparticles were dispersed in the solution to form a stable colloidal solution. In the revised manuscript, this sentence has been changed to “Then, 200 mg of F-127 and 200 mg of dopamine hydrochloride were added and stirred to form a stable white colloidal solution.”.

Comments 2. *The calculated BET surface area of ZIF-8 in p-ANHs is 1086 m²g⁻¹, the authors thought it was “nearly to that of the pristine ZIF-8 nanoparticles (Table S1)”. However, the BET surface area of pristine ZIF-8 was 1457 m²g⁻¹ (see Table S1). There is a 27% large difference and therefore the authors should give a reasonable explanation.*

Answer: Thanks very much for the comment. We have accepted it. In the revised manuscript, we have provided an explanation about the decrease of surface area of ZIF-8. We speculate that the growth of mPDA on the {100} facets of pristine ZIF-8 may cause the blockage of micropores on the {100} facets of the pristine ZIF-8, which further result in the decrease of the specific surface area. In addition, the defects at the interface of mPDA and ZIF-8 may also induce the decrease of surface area of pristine ZIF-8.

Correspondingly, we have added the following description on page 5 of the revised manuscript:

Based on this, the calculated net Brunauer-Emmett-Teller (BET) surface area of ZIF-8 building blocks in type 1 p-ANHs is 1086 m²/g, which is smaller than the net surface area of the pristine ZIF-8 nanoparticles (1457 m²/g) (Table S1). We speculate that the growth of mPDA on the {100} facets of pristine ZIF-8 may cause the blockage of micropores on the {100} facets of the pristine ZIF-8, which further result in the decrease of the specific surface of the micropores. In addition, the defects at the interface of mPDA and ZIF-8 may also induce the decrease of surface area of pristine ZIF-8.

Comments 3. *ZIF-8 has excellent chemical stability but is prone to degrade under acidic conditions. The {100} facet and {110} facet were degraded to different degrees as the dopamine hydrochloride was added. It seems that Zn was detected in the type 1 p-ANHs. Is It possible the defects formed and induced the anisotropic growth of mPDA?*

Answer: Thank you very much for the useful comments and question. We agree with the reviewer that there may be defects at the interface of mPDA and ZIF-8, because of the presence of Zn in the

mPDA subunits. However, we consider that the defects are not induced by the acidic environment, but the strong coordination effect of dopamine, since our reaction conditions are in alkaline conditions (~pH 10). The Zn doped mPDA subunits is present in both type 1 and type 2 architectures, therefore, we believe that the presence of defects is not the essential cause of selective growth, but the difference in binding energy of dopamine on different facets of crystalline ZIF-8.

Correspondingly, we have added the following description on Page 4 of the revised manuscript: Zn element can also be clearly observed in the mPDA subunits, indicating that the Zn^{2+} in the pristine ZIF-8 migrated from the MOF framework to the mPDA subunits. We speculate that this migration is induced by the strong coordination effect of dopamine with Zn^{2+} , and this migration may cause some defects at the interface of mPDA and ZIF-8.

Comments 4. *For ZIF-8, the {110} facet contains six-membered rings in the surface plane. Please check Figure 4A on page 14.*

Answer: Thanks very much for this useful comment. We have accepted it. We have made a revision in the revised manuscript.

Correspondingly, we have revised Figure 4 on Page 15 of the revised manuscript:

Figure 4. The mechanism of selective occupation strategy. (A) Schematic illustration of the coordination environments of the Zn^{2+} at {100} and {110} facets of ZIF-8. (B) Density Functional Theory (DFT) simulation results of the binding energy of the dopamine oligomer (DHI-dimer) and TMP occupation molecule on {100} and {110} facets of ZIF-8.

Comments 5. *On page4 “the TRD ZIF-8 nanoparticles are used as a host for the deposition of dopamine precursors under alkaline condition”, the pH value should be indicated.*

Answer: We thank the reviewer very much for the valuable comment. The pH value has been added in the revised supplementary information.

Comments 6. *In the Methods section, the Full name of Tris should be marked before the abbreviation used in “Tris aqueous solution”.*

Answer: We thank the reviewer for the useful comments. The full name “Tris(hydroxymethyl) aminomethane” has been added in the methods section.

Comments 7. *In conclusion section, the authors emphasized they “achieve f site-specific anisotropic growth”, what was the “f site”? they did not mentioned it in main text of the manuscript. Besides, “The research “boundary” between the crystalline p-ANHs and amorphous p-ANHs is crossed” and “The obtained various p-ANHs with intricate and unprecedented architectures across the research boundary between crystalline p-ANHs and amorphous p-ANHs”, were the two sentences repeated?*

Answer: We thank the reviewer very much. We are sorry for the misunderstanding. The "f" is a typo. This sentence has been changed to “In summary, we have demonstrated a selective occupation strategy to achieve site-specific anisotropic growth of amorphous mesoporous polydopamine (mPDA) on crystalline ZIF-8.”. Besides, we have deleted the sentence “The research “boundary” between the crystalline p-ANHs and amorphous p-ANHs is crossed” in the conclusion section.

REVIEWERS' COMMENTS

Reviewer #1 (Remarks to the Author):

The authors have addressed all my questions. I think the manuscript is publishable.

Reviewer #2 (Remarks to the Author):

I had the same impression as the first one on the revised manuscript. The facet selective modification of single crystalline particles by the specific molecules is well-established as in the authors response. The assembly of the modified nano or macro particles is also reported by many previous works. As I commented before, if the important finding of the present work is the use of multiple scale pores for the drug delivery, the manuscript fit more for the publication in the specific journals.

In addition, authors called the resultant particles as "anisotropic nanohybrid". I think this is misleading. Obtained particles seems to have centrosymmetric structures. (One of the resultant particles exhibited cubic structure as in the manuscript) I do not understand what "anisotropic" stands for. We usually do not use "anisotropic or anisotropy growth" for the surface grafting of the particles by organic molecules.

Accordingly, I recommend the manuscript should be forwarded to "Communications Chemistry" or other journals targeting more specific topics.

Reviewer #3 (Remarks to the Author):

In this revised manuscript, the authors have well addressed my concerns by point-by-point response and have improved their manuscript accordingly. I think it can be published in Nature Communications as it is, no further modification is needed.

Point-by-Point Response to Referees

Reviewer #1:

The authors have addressed all my questions. I think the manuscript is publishable.

Answer: Many thanks to the reviewer for their useful comments and suggestions during the peer review process.

Reviewer #2:

I had the same impression as the first one on the revised manuscript. The facet selective modification of single crystalline particles by the specific molecules is well-established as in the authors response. The assembly of the modified nano or macro particles is also reported by many previous works. As I commented before, if the important finding of the present work is the use of multiple scale pores for the drug delivery, the manuscript fit more for the publication in the specific journals.

In addition, authors called the resultant particles as “anisotropic nanohybrid”. I think this is misleading. Obtained particles seems to have centrosymmetric structures. (One of the resultant particles exhibited cubic structure as in the manuscript) I do not understand what “anisotropic” stands for. We usually do not use “anisotropic or anisotropy growth” for the surface grafting of the particles by organic molecules.

Accordingly, I recommend the manuscript should be forwarded to “Communications Chemistry” or other journals targeting more specific topics.

Answer: We thank the reviewer very much for the comments. We agree with the reviewer that the facet selective modification of single crystalline particles by the specific molecules is well-established, and the nanocrystal-based anisotropic nanohybrids are also reported by many previous works. However, different from the nanocrystal-based anisotropic nanohybrids, this work focuses on the controllable synthesis of porous anisotropic nanohybrids (p-ANHs). This is the first report on the site-specific anisotropic assembly of amorphous mesoporous subunits on crystalline metal–organic framework. The p-ANHs can not only provide multiple independent and exposed surface chemistries for the enhanced matter/energy exchange efficiency with external environments, but also possess high surface area, tunable pore sizes and structures, controllable framework compositions for further site-specific loading/grafting of functional entities. These features make p-ANHs have great application prospects in multi-guests co-delivery, nanomedicine, biphasic cascade catalysis, photocatalysis and so on. So, we believe that this work may appeal to the broad readership not only in the nanomaterials synthesis, but also in the applications of porous nanohybrids, such as drug delivery, catalysis, energy conversion and storage etc.

In contrast to isotropic growth (which tends to form core-shell structure), the anisotropic growth in this work means that amorphous mPDA only grows on the partial surface of the crystalline MOF. So, unlike the core-shell structured nanohybrids with isotropic surface chemistry of shell, one of the most important features of the obtained p-ANHs is that they can provide multiple independent and exposed surface chemistries, because the surfaces of both MOF and mPDA are exposed. Although the term “anisotropic or anisotropy” is not commonly used for the surface grafting of the particles by organic molecules, it is often used in the structural modulation of nanohybrids. Here we only list a few Review papers as examples, such as *Coord. Chem. Rev.*, 2021, 432, 213743; *Chem. Soc. Rev.*, 2020, 49, 1955; *Chem. Rev.*, 2019, 119, 12208; *Nano Today*, 2011, 6, 286; *Chem. Soc. Rev.*, 2011, 40, 2402 etc. By the way, for type 1 and 2 nanohybrids, they are not cubic structure. Accordingly, the manuscript has been revised to make the meaning of “anisotropic” clearer.

We have added the following description on Page 8 of the revised manuscript: Unlike the core-shell structured nanohybrids obtained by isotropic growth/assembly strategy (which can only exhibit isotropic surface chemistry of shell), one of the most important features of the obtained p-ANHs is that they can not only provide multiple independent and exposed surface chemistries for the enhanced matter/energy exchange

efficiency with external environments, but also possess multiple surfaces/storage-spaces for site-specific loading/grafting of multiple functional entities.

Reviewer #3:

In this revised manuscript, the authors have well addressed my concerns by point-by-point response and have improved their manuscript accordingly. I think it can be published in Nature Communications as it is, no further modification is needed.

Answer: Many thanks to the reviewer for their useful comments and suggestions during the peer review process.